# Distribution Aligning Refinery of Pseudo-label for Imbalanced Semi-supervised Learning

**Jaehyung Kim**[1], **Youngbum Hur**[2], **Sejun Park**[1],
**Eunho Yang**[1,3], **Sung Ju Hwang**[1,3], **Jinwoo Shin**[1]
[1]Korea Advanced Institute of Science and Technology (KAIST)
[2]Samsung Advanced Institute of Technology
[3]AItrics
{jaehyungkim,sejun.park,sjhwang82,jinwoos}@kaist.ac.kr
youngbum.hur@samsung.com yangeh@gmail.com

## Abstract

While semi-supervised learning (SSL) has proven to be a promising way for leveraging unlabeled data when labeled data is scarce, the existing SSL algorithms typically assume that training class distributions are balanced. However, these SSL algorithms trained under imbalanced class distributions can severely suffer when generalizing to a balanced testing criterion, since they utilize biased pseudo-labels of unlabeled data toward majority classes. To alleviate this issue, we formulate a convex optimization problem to softly refine the pseudo-labels generated from a biased model, and develop a simple iterative algorithm, named Distribution Aligning Refinery of Pseudo-label (DARP) that solves it provably and efficiently. Under various class-imbalanced semi-supervised scenarios, we demonstrate the effectiveness of DARP and its compatibility with state-of-the-art SSL schemes.

## 1   Introduction

It has been repeatedly shown that deep neural networks (DNNs) can achieve human- or super-human-level performances on various tasks [1, 17, 32]. This success, however, crucially relies on the availability of large-scale labeled datasets, which typically requires a lot of human efforts to be constructed. For example, the cost for labeling sequential (such as video and speech) data is often proportional to their lengths. Furthermore, some specific domain knowledge is often critical for labeling (such as medical) data. Semi-supervised learning (SSL) is one of promising, conventional ways to bypass this cost by leveraging unlabeled data for improving the performance of DNNs, given a small amount of labeled data [4, 5, 42]. The common approach of these modern state-of-the-art SSL algorithms is producing pseudo-labels for unlabeled data based on a model's prediction and then utilize the generated pseudo-labels for training the model iteratively [28, 36].

Most previous works on the line usually assume a balanced class distribution for both labeled and unlabeled datasets. However, in many realistic scenarios, the underlying class distribution of training data is highly imbalanced [27, 37]. It is well known that such an imbalanced class distribution hurts the generalization of DNNs, i.e., makes their predictions to be biased toward majority classes [13]. In other words, DNNs trained under an imbalanced class distribution suffer when generalizing to a balanced testing criterion. This issue can be more problematic for SSL algorithms since they generate pseudo-labels of unlabeled data from the model's biased predictions, i.e., pseudo-labels are even more severely imbalanced compared to true labels of labeled or unlabeled data. For example, when we train a Wide ResNet [43] on CIFAR-10 [12] under the imbalance ratio $\gamma = 150$[1] using a recent

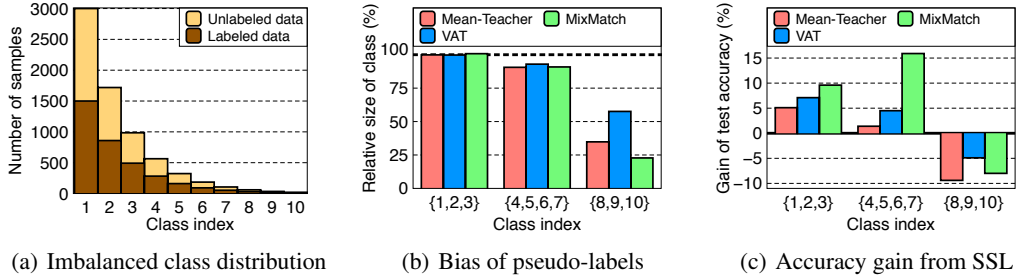

(a) Imbalanced class distribution     (b) Bias of pseudo-labels     (c) Accuracy gain from SSL

Figure 1: Experimental results on CIFAR-10 under the imbalance ratio $\gamma = 150$. (a) Class distribution of labeled and unlabeled data. (b) Relative size of pseudo-labels compares to size of true labels for each SSL algorithm. It is noticeable that the models fail to generate pseudo-labels on minority classes; hence the imbalance ratio of pseudo-labels is much larger than a true ratio $\gamma = 150$. (c) Test accuracy gain due to SSL algorithms compares to the vanilla model trained using only labeled data.

SSL algorithm, MixMatch [5], the resulting imbalance ratio of pseudo-labels becomes $\gamma = 1046$, which is much larger than the true ratio $\gamma = 150$ (see Figure 1(b) for detailed class-wise statistics).

Due to the aforementioned reason, we found that the performance of classifiers trained by recent SSL algorithms under the imbalanced class distribution often degrades on minority classes, even compared to the vanilla scheme using only labeled training samples (see Figure 1(c)). This implies that utilizing imbalanced unlabeled data for training can be dangerous for the classes having relatively small number of samples. Identifying a potential risk under class-imbalanced SSL scenarios is an important but under-explored research problem, up to date.

**Contribution.** To handle the issue, we propose a simple technique, coined distribution aligning refinery of pseudo-label (DARP), applicable to any existing SSL scheme utilizing pseudo-labels of unlabeled data. Our high-level idea is to refine the original, biased pseudo-labels so that their distribution can match the true class distribution of unlabeled data. Importantly, we also constrain our refined pseudo-labels to be not too far from the original pseudo-labels (constructed by an SSL algorithm). Without such a constraint, the individual quality of refined pseudo-labels can be poor, even when their overall distribution matches the true distribution. Motivated by the insight, we formulate an optimization problem for constructing refined pseudo-labels: minimizing the distortion from the original pseudo-labels, while matching the true class distribution.

The DARP algorithm is an efficient, iterative procedure to solve the proposed (convex) optimization with a provable guarantee. It finds the unique optimal solution by solving the Lagrangian dual of the original optimization. To further enhance the quality of the refined pseudo-labels, we additionally suggest removing some small and noisy entries in the original pseudo-labels when running DARP.

We demonstrate the effectiveness of the proposed approach under various realistic scenarios by varying the imbalanced class distributions. Despite its simplicity, the proposed DARP algorithm improves recent state-of-the-art SSL algorithms in all test cases, e.g., our method improves MixMatch [5], ReMixMatch [4] and FixMatch [35] with up to 77.2%, 31.4% and 53.1% relative reduction on the balanced test error, respectively. As expected, we find that our method is more effective when the (class) distribution mismatch between labeled and unlabeled data becomes severe. We believe that DARP method can be a strong baseline when other researchers pursue the related tasks in the future.

## 2 Related works

**Learning with class-imbalanced data.** Despite having several well-organized datasets in research, e.g., CIFAR [23] and ILSVRC [33], real-world datasets usually have a "long-tailed" label distribution [27, 37]. It is well-known that such *class-imbalanced* datasets make the standard training of DNN harder to generalize [13, 31, 39]. A natural approach to bypass this *class-imbalance problem* is re-balancing the training objective with respect to the class-wise sample sizes. Two of such methods are representative: (a) *re-weighting* the given loss function by a factor inversely proportional to the sample frequency in a class-wise manner [19, 22], and (b) *re-sampling* the given dataset so that the expected sampling distribution during training can be balanced [9, 16, 20]. However, naïvely

re-balancing the objective usually results in harsh over-fitting to minority classes, so several attempts have been made to alleviate this issue: [12] proposed the concept of "effective number" for each class as alternative weights in the re-weighting method. [8, 21] found that both re-weighting and re-sampling can be much more effective when applied at the later stage of training, in case of DNNs. Recently, [26] suggested re-balancing the training objective using unlabeled data, but it performs much worse compared to recent state-of-the-art semi-supervised learning algorithms.

**Semi-supervised learning.** The goal of a semi-supervised learning (SSL) algorithm is to improve the model's performance by leveraging unlabeled data to alleviate the need for labeled data. A popular class of SSL algorithms can be roughly viewed as producing a pseudo-label for each unlabeled data based on the model's prediction and then training the model to predict the pseudo-label when the unlabeled data is given as input. For example, pseudo labeling [25] (also known as self-training [41, 42]) generates the pseudo-label using the model's class prediction and trains the model with it again. Similarly, consistency regularization based methods obtain pseudo-label utilizing the model's predicted distribution after arbitrarily modifying the input [28, 34] or model function [36]. Recent state-of-the-art SSL algorithms combine both schemes for producing better pseudo-labels [4, 5]. However, since the pseudo-label of the unlabeled data is generated from the model's prediction, these methods can be inefficient or even harmful when the model's prediction is biased toward majority classes due to the imbalanced class distribution (see Figure 1(c)).

## 3 Handling imbalanced semi-supervised learning

### 3.1 Pseudo-label under imbalanced semi-supervised learning

We first describe the problem setup of our interest. Consider a classification problem with $K$ classes. Let $x \in \mathbb{R}^d$ and $y \in \{0, 1\}^K$ denote an input vector and corresponding one-hot label, respectively, where $d$ is the dimension of the input. We assume the following datasets are available:

$$\mathcal{D}^{\texttt{labeled}} = \left\{ (x_n^{\texttt{labeled}}, y_n^{\texttt{labeled}}) \right\}_{n=1}^N, \ \mathcal{D}^{\texttt{unlabeled}} = \left\{ x_m^{\texttt{unlabeled}} \right\}_{m=1}^M,$$

where $\mathcal{D}^{\texttt{labeled}}, \mathcal{D}^{\texttt{unlabeled}}$ correspond to labeled and unlabeled datasets, respectively. Then, the goal of the learner is to train a classifier $f = [f_k]_{k=1}^K : \mathbb{R}^d \to [0, 1]^K$ using the above datasets: it outputs the predictive probability $f_k(x) \in [0, 1]$ for each class $k$ given an input $x$. We also let $y_m^{\texttt{unlabeled}}$ denote the one-hot label of the true (yet unknown) class of $x_m^{\texttt{unlabeled}}$. The numbers of data in class $k$ under $\mathcal{D}^{\texttt{labeled}}$ and $\mathcal{D}^{\texttt{unlabeled}}$ are denoted by $N_k$ and $M_k$, respectively, i.e., $\sum_{k=1}^K N_k = N$ and $\sum_{k=1}^K M_k = M$. We are interested in class-imbalanced scenarios where $\frac{\max_k N_k}{\min_k N_k}$ and $\frac{\max_k M_k}{\min_k M_k}$ are much larger than 1 as illustrated in Figure 1(a).

To utilize the unlabeled training data effectively, most recent state-of-the-art SSL algorithms infer their labels by some pseudo-labels (e.g., classifier's prediction of augmented data [4, 5]) denoted by

$$\left\{ \hat{y}_m^{\texttt{unlabeled}} \in [0, 1]^K \ \Big| \ \sum_{k=1}^K \hat{y}_m^{\texttt{unlabeled}}(k) = 1 \right\}_{m=1}^M.$$

Then, they train the model by optimizing supervised losses (e.g., cross-entropy) corresponding to $(x_n^{\texttt{labeled}}, y_n^{\texttt{labeled}})$ and $(x_m^{\texttt{unlabeled}}, \hat{y}_m^{\texttt{unlabeled}})$, possibly with other regularization losses, e.g., consistency loss [28, 36]. Hence, the performance of SSL algorithm is quite sensitive to the quality of pseudo-labels $\hat{y}_m^{\texttt{unlabel}}$. However, the imbalanced class distribution incurs the bias of the model's prediction toward majority classes of large $N_k$, and the resulting quality of the pseudo-labels can be significantly degraded. As reported in Figure 1(b), pseudo-labels can be more severely imbalanced than the truth. Hence, some SSL algorithms utilizing these pseudo-labels as direct supervision or a source of regularization can be ineffective or even harmful for minority classes (see Figure 1(c)).

### 3.2 Distribution aligning refinery of pseudo-label

Now, we present our technique, coined distribution aligning refinery of pseudo-label (DARP). The input of DARP could be any pseudo-labels constructed from any SSL algorithm, i.e., it can incorporate into various SSL algorithms for refining their outputs. In this section, we focus on describing how to refine pseudo-labels given the true class distribution of unlabeled data, i.e., $\{M_k\}_{k=1}^K$. As $\{M_k\}_{k=1}^K$ is not known for the learner in general, we will discuss how to estimate it in Section 3.3.

---

**Algorithm 1** `DualCoordinateAscent`: Coordinate ascent algorithm for dual of (1)

---

**Require:** $\{\hat{y}_m^{\mathtt{unlabeled}}\}_{m=1}^M$, $\{w_m\}_{m=1}^M$, $\{M_k\}_{k=1}^K$, $T$
**Ensure:** The unique solution of (1)

---

1: $\hat{y}_m^0 \leftarrow y_m^{\mathtt{unlabeled}}$, $\alpha_m^0 \leftarrow 1$, $\beta_k^0 \leftarrow 1$, $\quad \forall m, k$
2: **for** $t = 1$ **to** $T$ **do**
3: $\quad$ **if** $t$ is odd or $t = T$ **then**
4: $\qquad \alpha_m^t \leftarrow \left( \sum_{k=1}^K \hat{y}_m^0(k)(\beta_k^{t-1})^{\frac{1}{w_m}} \right)^{-1}, \quad \forall m,$
5: $\qquad \beta_k^t \leftarrow \beta_k^{t-1}, \quad \forall k$
6: $\quad$ **else**
7: $\qquad \alpha_m^t \leftarrow \alpha_m^{t-1}, \quad \forall m$
8: $\qquad \beta_k^t \leftarrow \mathtt{Solve}_{Z \geq 0} \left( \sum_{m=1}^M \hat{y}_m^0(k)\alpha_m^{t-1} Z^{\frac{1}{w_m}} - M_k \right), \quad \forall k$ [2]
9: $\quad$ **end if**
10: **end for**
11: $\hat{y}_m^{\mathtt{out}}(k) \leftarrow \hat{y}_m^0(k)\alpha_m^T(\beta_k^T)^{\frac{1}{w_m}}, \quad \forall m, k$

---

**Refining pseudo-labels via optimization.** Given the original pseudo-labels $\{\hat{y}_m^{\mathtt{unlabeled}}\}_{m=1}^M$ (generated by an SSL algorithm), we are interested in refining them so that their class distribution matches the true distribution $\{M_k\}_{k=1}^K$. Simultaneously, we also want to preserve the original information in $\{\hat{y}_m^{\mathtt{unlabeled}}\}_{m=1}^M$ as much as possible, to maintain the high quality of refined pseudo-labels. To this end, we propose the following convex optimization problem with respect to variables $\{\hat{y}_m\}_{m=1}^M$:

$$\text{minimize} \quad \sum_{m=1}^M w_m D_{KL}(\hat{y}_m \,||\, \hat{y}_m^{\mathtt{unlabeled}}) \tag{1}$$

$$\text{subject to} \quad \sum_{m=1}^M \hat{y}_m(k) = M_k, \; \forall k, \; \sum_{k=1}^K \hat{y}_m(k) = 1, \; \forall m, \; \hat{y}_m(k) \in [0,1], \; \forall m, k$$

where the KL-divergence objective $D_{KL}(\hat{y}_m \,||\, \hat{y}_m^{\mathtt{unlabeled}})$ is to preserve the original information of $\hat{y}_m^{\mathtt{unlabeled}}$ and the constraint $\sum_{m=1}^M \hat{y}_m(k) = M_k$ is to match to the true class distribution of unlabeled data. In particular, the above optimization is encouraged to preserve more information of high-confident original pseudo-labels by introducing weight $w_m$ to each data by

$$w_m := \left( H(\hat{y}_m^{\mathtt{unlabeled}}) \right)^{-1},$$

i.e., larger weight to more confident data of smaller entropy $H$.

Solving the optimization (1), e.g., by some generic convex scheme, might incur much computational overhead, especially for refining a large number of pseudo-labels. To address this issue, we propose an efficient iterative procedure, Algorithm 1, for solving (1). In essence, it is a coordinate ascent algorithm for Lagrangian dual of (1), which alternatively finds the local optimum of each of dual variables $\{\alpha_m\}_{m=1}^M$ and $\{\beta_k\}_{k=1}^K$. We provide the following provable guarantee of Algorithm 1.

**Theorem 1.** *The output of Algorithm 1 converges to the unique solution of (1) as $T \to \infty$ unless $\sum_{m=1}^M w_m D_{KL}(\hat{y}_m \,||\, \hat{y}_m^{\mathtt{unlabeled}}) = \infty$ for all feasible $\{\hat{y}_m\}_{m=1}^M$.*

We present the formal derivation of Algorithm 1 and the proof of Theorem 1 in Section A and B of the supplementary material. In our experiments, we empirically observe that $T = 10$ is enough for the convergence.[3]

**Removing small entries of pseudo-labels.** To further enhance the quality of pseudo-labels, we would concentrate on confident entries of the original pseudo-labels by removing small and noisy entries as below:

$$\hat{y}_m^0(k) \leftarrow \begin{cases} \hat{y}_m^{\mathtt{unlabeled}}(k) & \text{if } x_m^{\mathtt{unlabeled}} \in \mathcal{U}_k \\ 0 & \text{otherwise} \end{cases}, \tag{2}$$

**Algorithm 2** DARP: Distribution aligning refinery of pseudo-label

---

**Input:** Unlabeled data $\{x_m^{\texttt{unlabeled}}\}_{m=1}^M$, pseudo-labels $\{\hat{y}_m^{\texttt{unlabeled}}\}_{m=1}^M$, sample-wise weights $\{w_m\}_{m=1}^M$, true class distribution $\{M_k\}_{k=1}^K$, number of classes $K$, number of iterations $T$, hyper-parameter for removing noisy pseudo-label entries $\delta$.
**Output:** Refined pseudo-labels $\hat{y}_m^{\texttt{DARP}}$

---

1: **for** $k = 1$ **to** $K$ **do**
2: $\quad \{\hat{m}_1, \dots, \hat{m}_M\} \leftarrow \texttt{Sort}\big(\{y_m^{\texttt{unlabeled}}(k) \,|\, \forall m\}\big)$
3: $\quad \mathcal{U}_k = \{x_{\hat{m}_1}^{\texttt{unlabeled}}, \dots, x_{\hat{m}_{\hat{\delta}}}^{\texttt{unlabeled}}\}, \hat{\delta} = \lfloor \delta \cdot M_k \rfloor$
4: **end for**
5: $\hat{y}_m^{\texttt{unlabeled}}(k) \leftarrow \begin{cases} \hat{y}_m^{\texttt{unlabeled}}(k) & \text{if } x_m^{\texttt{unlabeled}} \in \mathcal{U}_k \\ 0 & \text{otherwise} \end{cases}, \; \alpha_m^0 = \beta_k^0 = 1 \quad \forall m, k$
6: $\hat{y}_m^{\texttt{DARP}}(k) \leftarrow \texttt{DualCoordinateAscent}\big(\{\hat{y}_m^{\texttt{unlabeled}}\}_{m=1}^M, \{w_m\}_{m=1}^M, \{M_k\}_{k=1}^K, T\big)$

---

where $\mathcal{U}_k$ is the subset of top $\delta \cdot M_k$ unlabeled data having larger values of $y_m^{\texttt{unlabeled}}(k)$ for some $\delta > 0$. Namely, we clip the value of $y_m^{\texttt{unlabeled}}(k)$ to be zero if it is relatively small. At some angle, the initialization (2) is expected to decrease the entropy of $\hat{y}_m^t$, compared to the naïve initialization $\hat{y}_m^0 = \hat{y}_m^{\texttt{unlabeled}}$, which is similar to the entropy minimization technique used in the previous SSL algorithm [15] in spirit. The full details of the proposed DARP are described in Algorithm 2.

### 3.3 Estimating class distribution of unlabeled data

Recall that DARP requires the true class distribution of unlabeled data, $\{M_k\}_{k=1}^K$. If both labeled and unlabeled data are sampled from the same distribution (arguably the most practical scenario), $\{M_k\}_{k=1}^K$ can be inferred from that of labeled data. Otherwise, we suggest to use the following simple procedure for estimating it using a confusion matrix $C^{\texttt{unlabeled}} \in \mathbb{R}^{K \times K}$:

$$\begin{bmatrix} M_1 \\ \vdots \\ M_K \end{bmatrix} = \big(C^{\texttt{unlabeled}}\big)^{-1} \times \begin{bmatrix} \sum_{m=1}^M f_1(x_m^{\texttt{unlabeled}}) \\ \vdots \\ \sum_{m=1}^M f_K(x_m^{\texttt{unlabeled}}) \end{bmatrix}, C_{ij}^{\texttt{unlabeled}} := \frac{\sum_{m:y_m^{\texttt{unlabeled}}(j)=1} f_i(x_m^{\texttt{unlabeled}})}{|\{m \,|\, y_m^{\texttt{unlabeled}}(j) = 1\}|}$$

where $C_{ij}^{\texttt{unlabeled}}$ denote the empirical probability that the model predicts class $i$ when the true class is $j$. The equation is derived from the definition of $C^{\texttt{unlabeled}}$ as follow:

$$\begin{bmatrix} \sum_{m=1}^M f_1(x_m^{\texttt{unlabeled}}) \\ \vdots \\ \sum_{m=1}^M f_K(x_m^{\texttt{unlabeled}}) \end{bmatrix} = C^{\texttt{unlabeled}} \times \begin{bmatrix} |\{m \,|\, y_m^{\texttt{unlabeled}}(1) = 1\}| \\ \vdots \\ |\{m \,|\, y_m^{\texttt{unlabeled}}(K) = 1\}| \end{bmatrix} = C^{\texttt{unlabeled}} \times \begin{bmatrix} M_1 \\ \vdots \\ M_K \end{bmatrix}$$

However, to obtain the confusion matrix $C^{\texttt{unlabeled}}$, the true labels for unlabeled dataset are required which do not exist. To circumvent this, we approximate it using the given labeled dataset $\mathcal{D}^{\texttt{labeled}}$.[4] This estimation assumes that confusion matrices of labeled and unlabeled datasets are similar and it holds when both datasets are constructed from the same input distribution (although their label distributions are different). Also, it is worth noting that similar approaches are used in the case of noisy labels [18] and domain adaptation [3].

## 4 Experiments

In this section, we evaluate our algorithm on various scenarios for imbalanced semi-supervised learning in classification problems. We first describe the experimental setups in Section 4.1. In Section 4.2, we present empirical evaluations on DARP and other baseline algorithms under various setups. In Section 4.3, we present detailed analysis on DARP.

### 4.1 Experimental setup

**Imbalanced dataset.** We consider "synthetically long-tailed" variants of CIFAR-10, CIFAR-100 [23], and STL-10 [11] in order to evaluate our algorithm under various levels of imbalance. Results on real-world dataset, SUN-397 [40], are also given in Section C of the supplementary material. For constructing the class-imbalanced training dataset, without loss of generality, we assume the ordered numbers of labeled data in each class as $N_1 \geq \cdots \geq N_K$. We use a single parameter $\gamma_l \geq 1$, called the *imbalance ratio*, to control the class-imbalance of the labeled dataset: once $\gamma_l$ and $N_1$ are given, we set $N_k = N_1 \cdot \gamma_l^{-\frac{k-1}{K-1}}$ so that $N_1 = \gamma_l \cdot N_K$ as done by [12]. Namely, larger $\gamma_l$ indicates more imbalanced class distribution. Likewise, we assume that $M_1 \geq \cdots \geq M_K$ for the unlabeled dataset and its class-imbalance is controlled by $\gamma_u \geq 1$, as we did for the labeled dataset. We use $N_1 = 1500, M_1 = 3000$ for CIFAR-10 and $N_1 = 150, M_1 = 300$ for CIFAR-100, respectively. Figure 1(a) illustrates the constructed imbalanced class distribution on CIFAR-10 with $\gamma_l = \gamma_u = 150$. To evaluate the classification performance of models trained under the imbalanced dataset, we report two popular metrics: *balanced accuracy* (bACC) [19, 39] and *geometric mean scores* (GM) [7, 24], which are defined by the arithmetic and geometric mean over class-wise sensitivity, respectively. In this section, mean and standard deviation are reported across three random trials, respectively.

**Baselines.** We compare our algorithm with various baselines, including recent re-balancing algorithms for learning with class-imbalanced labeled data only (i.e., *without using unlabeled data*) and semi-supervised learning algorithms for learning with both labeled and unlabeled data (i.e., *without considering class-imbalance*). We first consider a naïve baseline without any re-balancing and using unlabeled data, denoted by (*a*) Vanilla. Then, we consider a wide range of previous "re-balancing" algorithms denoted by (*b*) Re-sampling [20]: each class is equally sampled for training; (*c*) Label-distribution-aware margin (LDAM-DRW) [8]: the classifier is trained to impose larger margin to minority classes and balancing the objective at the later stage of training; (*d*) Classifier re-training (cRT) [21]: only re-train the classifier with the balanced objective after training a whole network under imbalanced distribution. We also consider a wide range of previous "semi-supervised learning" algorithms denoted by (*e*) Virtual adversarial training (VAT) [28]: for unlabeled data, consistency regularization with its adversarial example is added; (*f*) Mean-Teacher [36]: adding consistency regularization between the prediction of the current model and the ensemble of previous models; (*g*) MixMatch [5]: both pseudo-label and consistency regularization are applied with Mixup regularization [44]; (*h*) ReMixMatch [4]: MixMatch is further improved with an augmentation anchoring and a distribution alignment. (*i*) FixMatch [35]: strongly augmented unlabeled data are used for training where their pseudo-labels are generated from their weakly augmented version. Details on the implementation of the baseline algorithms are presented in Section E of the supplementary material.

**Training details.** All experiments are conducted with Wide ResNet-28-2 [30] and it is trained with batch size 64 for $2.5 \times 10^5$ training iterations. For all algorithms, we evaluate the model on the test dataset for every 500 iterations and report the average test accuracy of the last 20 evaluations following [5]. We apply the DARP procedure for every 10 iterations with fixed hyper-parameters $\delta = 2$ and $T = 10$, which is empirically enough for the convergence of DARP. Since pseudo-labels are not accurate at the early stage of training, we are not using DARP until the first 40% of iterations. More training details are presented in Section E of the supplementary material.

### 4.2 Imbalanced semi-supervised learning

We evaluate DARP with both re-balancing (RB) and semi-supervised learning (SSL) algorithms under various levels of class-imbalance. We apply DARP to recent state-of-the-art SSL algorithms: Mix-Match [5], ReMixMatch [4] and FixMatch [35], denoted by MixMatch+DARP, ReMixMatch+DARP and FixMatch+DARP, respectively, and observe the gain due to DARP.

**CIFAR-10 under $\gamma_l = \gamma_u$.** We first conduct experiments in the case $\gamma := \gamma_l = \gamma_u$. This is arguably the most natural scenario that each data in both datasets is sampled from the same distribution. Here, we choose $M_k \propto N_k$ for both DARP and ReMixMatch. To verify the effectiveness of DARP in this scenario, we compare DARP with various semi-supervised learning (SSL) algorithms and re-balancing (RB) algorithms on CIFAR-10 with various $\gamma$. Table 1 summarizes the performance of baseline algorithms with/without DARP for learning CIFAR-10. It is noticeable that many SSL algorithms perform worse than RB algorithms, even they use more training (unlabeled) data. We observe that this is because the pseudo-labels of SSL algorithms generated from the biased models are likely to follow the majority classes of labeled data (see Figure 2(a)). Hence, utilizing these

Table 1: Comparison of classification performance (bACC/GM) on CIFAR-10 under three different class-imbalance ratio $\gamma = \gamma_l = \gamma_u$. SSL denotes semi-supervised learning and RB denotes re-balancing. The numbers in brackets below the gray rows are relative test error gains from DARP, compared to applied baseline SSL algorithms, respectively. The best results are indicated in bold.

| | | | CIFAR-10 ($\gamma = \gamma_l = \gamma_u$) | | |
|---|---|---|---|---|---|
| Algorithm | SSL | RB | $\gamma = 50$ | $\gamma = 100$ | $\gamma = 150$ |
| Vanilla | - | - | $65.2_{\pm0.05}$ / $61.1_{\pm0.09}$ | $58.8_{\pm0.13}$ / $51.0_{\pm0.11}$ | $55.6_{\pm0.43}$ / $44.0_{\pm0.98}$ |
| Re-sampling [20] | - | ✓ | $64.3_{\pm0.48}$ / $60.6_{\pm0.67}$ | $55.8_{\pm0.47}$ / $45.1_{\pm0.30}$ | $52.2_{\pm0.05}$ / $38.2_{\pm1.49}$ |
| LDAM-DRW [8] | - | ✓ | $68.9_{\pm0.07}$ / $67.0_{\pm0.08}$ | $62.8_{\pm0.17}$ / $58.9_{\pm0.60}$ | $57.9_{\pm0.20}$ / $50.4_{\pm0.30}$ |
| cRT [21] | - | ✓ | $67.8_{\pm0.13}$ / $66.3_{\pm0.15}$ | $63.2_{\pm0.45}$ / $59.9_{\pm0.40}$ | $59.3_{\pm0.10}$ / $54.6_{\pm0.72}$ |
| VAT [28] | ✓ | - | $70.6_{\pm0.29}$ / $67.8_{\pm0.19}$ | $62.6_{\pm0.40}$ / $55.1_{\pm0.56}$ | $57.9_{\pm0.42}$ / $46.3_{\pm0.47}$ |
| Mean-Teacher [36] | ✓ | - | $68.8_{\pm1.05}$ / $64.9_{\pm1.53}$ | $60.9_{\pm0.33}$ / $52.8_{\pm0.81}$ | $54.5_{\pm0.22}$ / $39.8_{\pm0.73}$ |
| MixMatch [5] | ✓ | - | $73.2_{\pm0.56}$ / $68.9_{\pm1.15}$ | $64.8_{\pm0.28}$ / $49.0_{\pm2.05}$ | $62.5_{\pm0.31}$ / $42.5_{\pm1.68}$ |
| MixMatch + DARP | ✓ | - | $75.2_{\pm0.47}$ / $72.8_{\pm0.63}$ | $67.9_{\pm0.14}$ / $61.2_{\pm0.15}$ | $65.8_{\pm0.52}$ / $56.5_{\pm2.08}$ |
| | | | (-7.41% / -12.6%) | (-8.77% / -23.8%) | (-8.69% / -24.4%) |
| ReMixMatch [4] | ✓ | - | $81.5_{\pm0.26}$ / $80.2_{\pm0.32}$ | $73.8_{\pm0.38}$ / $69.5_{\pm0.84}$ | $69.9_{\pm0.47}$ / $62.5_{\pm0.35}$ |
| ReMixMatch + DARP | ✓ | - | $\mathbf{82.1}_{\pm0.14}$ / $80.8_{\pm0.09}$ | $\mathbf{75.8}_{\pm0.09}$ / $72.6_{\pm0.24}$ | $\mathbf{71.0}_{\pm0.27}$ / $64.5_{\pm0.68}$ |
| | | | (-3.45% / -3.52%) | (-7.84% / -10.2%) | (-3.60% / -5.19%) |
| FixMatch [35] | ✓ | - | $79.2_{\pm0.33}$ / $77.8_{\pm0.36}$ | $71.5_{\pm0.72}$ / $66.8_{\pm1.51}$ | $68.4_{\pm0.15}$ / $59.9_{\pm0.43}$ |
| FixMatch + DARP | ✓ | - | $81.8_{\pm0.24}$ / $\mathbf{80.9}_{\pm0.28}$ | $75.5_{\pm0.05}$ / $\mathbf{73.0}_{\pm0.09}$ | $70.4_{\pm0.25}$ / $\mathbf{64.9}_{\pm0.17}$ |
| | | | (-12.9% / -14.1%) | (-14.0% / -18.8%) | (-22.4% / -20.3%) |

Table 2: Comparison of classification performance (bACC/GM) on CIFAR-10 under four different class-imbalance ratio $\gamma_u$ with $\gamma_l = 100$. SSL denotes semi-supervised learning and RB denotes re-balancing. The numbers in brackets below the gray rows are relative test error gains from DARP, compared to applied baseline SSL algorithms, respectively. The best results are indicated in bold.

| | | | CIFAR-10 ($\gamma_l = 100$) | | | |
|---|---|---|---|---|---|---|
| Algorithm | SSL | RB | $\gamma_u = 1$ | $\gamma_u = 50$ | $\gamma_u = 150$ | $\gamma_u = 100$ (reversed) |
| Vanilla | - | - | $58.8_{\pm0.13}$ / $51.0_{\pm0.11}$ | $58.8_{\pm0.13}$ / $51.0_{\pm0.11}$ | $58.8_{\pm0.13}$ / $51.0_{\pm0.11}$ | $58.8_{\pm0.13}$ / $51.0_{\pm0.11}$ |
| Re-sampling [20] | - | ✓ | $55.8_{\pm0.47}$ / $45.1_{\pm0.30}$ | $55.8_{\pm0.47}$ / $45.1_{\pm0.30}$ | $55.8_{\pm0.47}$ / $45.1_{\pm0.30}$ | $55.8_{\pm0.47}$ / $45.1_{\pm0.30}$ |
| LDAM-DRW [8] | - | ✓ | $62.8_{\pm0.17}$ / $58.9_{\pm0.60}$ | $62.8_{\pm0.17}$ / $58.9_{\pm0.60}$ | $62.8_{\pm0.17}$ / $58.9_{\pm0.60}$ | $62.8_{\pm0.17}$ / $58.9_{\pm0.60}$ |
| cRT [21] | - | ✓ | $63.2_{\pm0.45}$ / $59.9_{\pm0.40}$ | $63.2_{\pm0.45}$ / $59.9_{\pm0.40}$ | $63.2_{\pm0.45}$ / $59.9_{\pm0.40}$ | $63.2_{\pm0.45}$ / $59.9_{\pm0.40}$ |
| VAT [28] | ✓ | - | $65.2_{\pm0.12}$ / $59.5_{\pm0.26}$ | $64.0_{\pm0.31}$ / $57.3_{\pm0.66}$ | $62.8_{\pm0.19}$ / $55.1_{\pm0.70}$ | $59.4_{\pm0.36}$ / $50.6_{\pm0.61}$ |
| Mean-Teacher [36] | ✓ | - | $73.9_{\pm1.19}$ / $71.7_{\pm1.42}$ | $61.2_{\pm0.51}$ / $53.5_{\pm0.84}$ | $59.7_{\pm0.50}$ / $50.0_{\pm1.61}$ | $61.0_{\pm0.82}$ / $56.4_{\pm1.64}$ |
| MixMatch [5] | ✓ | - | $41.5_{\pm0.76}$ / $12.0_{\pm1.34}$ | $64.1_{\pm0.58}$ / $48.3_{\pm0.70}$ | $65.5_{\pm0.64}$ / $51.1_{\pm2.41}$ | $47.9_{\pm0.09}$ / $20.5_{\pm0.85}$ |
| MixMatch + DARP | ✓ | - | $86.7_{\pm0.80}$ / $86.2_{\pm0.82}$ | $68.3_{\pm0.47}$ / $62.2_{\pm1.21}$ | $66.7_{\pm0.25}$ / $58.8_{\pm0.42}$ | $72.9_{\pm0.24}$ / $71.0_{\pm0.32}$ |
| | | | (-77.2% / -84.4%) | (-11.8% / -27.0%) | (-3.62% / -15.7%) | (-48.0% / -63.6%) |
| ReMixMatch [4] | ✓ | - | $48.3_{\pm0.14}$ / $19.5_{\pm0.85}$ | $75.1_{\pm0.43}$ / $71.9_{\pm0.77}$ | $72.5_{\pm0.10}$ / $68.2_{\pm0.32}$ | $49.0_{\pm0.55}$ / $17.1_{\pm1.48}$ |
| ReMixMatch* | ✓ | - | $85.0_{\pm1.35}$ / $84.3_{\pm1.55}$ | $77.0_{\pm0.12}$ / $74.7_{\pm0.04}$ | $72.8_{\pm0.10}$ / $68.8_{\pm0.21}$ | $75.3_{\pm0.03}$ / $72.3_{\pm0.04}$ |
| ReMixMatch* + DARP | ✓ | - | $\mathbf{89.7}_{\pm0.15}$ / $\mathbf{89.4}_{\pm0.17}$ | $\mathbf{77.4}_{\pm0.22}$ / $75.0_{\pm0.25}$ | $\mathbf{73.2}_{\pm0.11}$ / $69.2_{\pm0.31}$ | $\mathbf{80.1}_{\pm0.11}$ / $\mathbf{78.5}_{\pm0.17}$ |
| | | | (-31.4% / -32.5%) | (-1.72% / -1.49%) | (-1.53% / -2.64%) | (-19.5% / -22.5%) |
| FixMatch [35] | ✓ | - | $68.9_{\pm1.95}$ / $42.8_{\pm8.11}$ | $73.9_{\pm0.25}$ / $70.5_{\pm0.52}$ | $69.6_{\pm0.60}$ / $62.6_{\pm1.11}$ | $65.5_{\pm0.05}$ / $26.0_{\pm0.44}$ |
| FixMatch + DARP | ✓ | - | $85.4_{\pm0.55}$ / $85.0_{\pm0.65}$ | $77.3_{\pm0.17}$ / $\mathbf{75.5}_{\pm0.21}$ | $72.9_{\pm0.24}$ / $\mathbf{69.5}_{\pm0.18}$ | $74.9_{\pm0.51}$ / $72.3_{\pm1.13}$ |
| | | | (-53.1% / -73.8%) | (-13.3% / -17.0%) | (-10.9% / -18.4%) | (-31.3% / -60.3%) |

biased pseudo-labels for training can be ineffective or even harmful. On the other hand, DARP refines such biased pseudo-labels correctly (see Figure 2(c)), and consequently, it improves the performance of all the applied SSL algorithms: MixMatch, ReMixMatch and FixMatch. For example, DARP exhibits 22.4%/20.3% relative error reductions of bACC/GM in the case of FixMatch under $\gamma = 150$. While DARP outperforms all the baselines, it could be even further improved by combining with RB algorithms (see Section D of the supplementary material).

**CIFAR-10 under $\gamma_l \neq \gamma_u$.** The imbalance ratio of unlabeled data may not be the same as that of labeled data in general, i.e., $\gamma_l \neq \gamma_u$ and $\gamma_u$ is unknown. In this case, we estimate $\{M_k\}_{k=1}^K$ for both ReMixMatch and DARP as described in Section 3.3.

Table 2 summarizes the experimental results under $\gamma_l \neq \gamma_u$. Here, we denote "ReMixMatch" for ReMixMatch without estimation of $\{M_k\}_{k=1}^K$, i.e., it assumes $M_k \propto N_k$, and "ReMixMatch*" for ReMixMatch with estimation of $\{M_k\}_{k=1}^K$. In Table 2, one can observe that DARP consistently improves all the baselines. Surprisingly, the relative error gain from DARP increases as $\gamma_u$ decreases,

Table 3: Comparison of classification performance (bACC/GM) on CIFAR-100 and STL-10 under two different class-imbalance ratio $\gamma_l$. SSL denotes semi-supervised learning and RB denotes re-balancing. The numbers in brackets below the gray rows are relative test error gains from DARP, compared to applied baseline SSL algorithms, respectively. The best results are indicated in bold.

| Algorithm | SSL | RB | CIFAR-100 ($\gamma_l = \gamma_u$) | | STL-10 ($\gamma_l \neq \gamma_u$) | |
|---|---|---|---|---|---|---|
| | | | $\gamma_l = 10$ | $\gamma_l = 20$ | $\gamma_l = 10$ | $\gamma_l = 20$ |
| Vanilla | - | - | $55.9_{\pm0.12}$ / $50.7_{\pm0.12}$ | $49.5_{\pm0.03}$ / $40.3_{\pm0.04}$ | $56.4_{\pm1.50}$ / $51.8_{\pm1.67}$ | $48.1_{\pm0.26}$ / $38.2_{\pm0.67}$ |
| Re-sampling | - | ✓ | $54.6_{\pm0.05}$ / $48.9_{\pm0.40}$ | $48.1_{\pm0.17}$ / $38.3_{\pm0.82}$ | $57.8_{\pm0.76}$ / $53.6_{\pm0.80}$ | $47.4_{\pm0.16}$ / $35.8_{\pm0.11}$ |
| LDAM-DRW | - | ✓ | $55.7_{\pm0.75}$ / $51.6_{\pm0.08}$ | $50.4_{\pm0.32}$ / $45.4_{\pm0.98}$ | $58.0_{\pm0.53}$ / $54.4_{\pm0.84}$ | $50.2_{\pm0.05}$ / $42.4_{\pm0.08}$ |
| cRT | - | ✓ | $56.2_{\pm0.36}$ / $52.2_{\pm0.38}$ | $50.7_{\pm0.11}$ / $43.8_{\pm0.04}$ | $59.2_{\pm0.53}$ / $55.7_{\pm0.65}$ | $49.2_{\pm0.29}$ / $42.3_{\pm0.20}$ |
| VAT | ✓ | - | $54.6_{\pm0.06}$ / $48.6_{\pm0.11}$ | $48.5_{\pm0.16}$ / $38.5_{\pm0.25}$ | $64.2_{\pm0.33}$ / $61.1_{\pm0.50}$ | $56.2_{\pm0.03}$ / $50.5_{\pm0.08}$ |
| Mean-Teacher | ✓ | - | $54.1_{\pm0.13}$ / $48.2_{\pm0.05}$ | $48.2_{\pm0.13}$ / $37.6_{\pm0.07}$ | $57.7_{\pm0.10}$ / $54.8_{\pm0.87}$ | $48.0_{\pm0.47}$ / $35.3_{\pm3.81}$ |
| MixMatch | ✓ | - | $60.1_{\pm0.39}$ / $48.1_{\pm4.08}$ | $53.4_{\pm0.04}$ / $41.9_{\pm0.16}$ | $56.3_{\pm0.46}$ / $48.2_{\pm1.08}$ | $45.2_{\pm0.19}$ / $22.0_{\pm0.12}$ |
| MixMatch + DARP | ✓ | - | $60.9_{\pm0.24}$ / $55.8_{\pm0.05}$ | $54.8_{\pm0.27}$ / $45.6_{\pm0.48}$ | $67.9_{\pm0.24}$ / $65.1_{\pm0.51}$ | $58.3_{\pm0.73}$ / $52.2_{\pm1.01}$ |
| | | | (-2.04% / -14.8%) | (-3.38% / -6.33%) | (-26.7% / -32.7%) | (-23.9% / -38.8%) |
| ReMixMatch | ✓ | - | $59.2_{\pm0.03}$ / $52.1_{\pm0.13}$ | $53.5_{\pm0.03}$ / $42.3_{\pm0.13}$ | $67.8_{\pm0.45}$ / $61.1_{\pm0.92}$ | $60.1_{\pm1.18}$ / $44.9_{\pm1.52}$ |
| ReMixMatch + DARP | ✓ | - | $59.8_{\pm0.20}$ / $52.9_{\pm0.41}$ | $54.4_{\pm0.07}$ / $44.2_{\pm0.07}$ | $\mathbf{79.4}_{\pm0.07}$ / $\mathbf{78.2}_{\pm0.10}$ | $\mathbf{70.9}_{\pm0.44}$ / $\mathbf{67.0}_{\pm1.62}$ |
| | | | (-1.25% / -1.59%) | (-1.88% / -3.23%) | (-36.0% / -44.0%) | (-27.0% / -40.0%) |
| FixMatch | ✓ | - | $60.1_{\pm0.05}$ / $54.4_{\pm0.11}$ | $54.0_{\pm0.04}$ / $44.4_{\pm0.17}$ | $72.9_{\pm0.09}$ / $69.6_{\pm0.01}$ | $63.4_{\pm0.21}$ / $52.6_{\pm0.09}$ |
| FixMatch + DARP | ✓ | - | $\mathbf{61.1}_{\pm0.23}$ / $\mathbf{56.4}_{\pm0.28}$ | $\mathbf{54.9}_{\pm0.05}$ / $\mathbf{46.4}_{\pm0.41}$ | $77.8_{\pm0.33}$ / $76.5_{\pm0.40}$ | $69.9_{\pm1.77}$ / $65.4_{\pm3.07}$ |
| | | | (-2.55% / -4.40%) | (-1.97% / -3.60%) | (-18.2% / -22.8%) | (-17.9% / -27.0%) |

i.e., the overall class-distribution becomes more balanced. We believe that this is because SSL algorithms without DARP cannot fully enjoy this more balanced distribution as their pseudo-labels of minority class data are significantly biased toward majority classes. Meanwhile, DARP correctly refines pseudo-labels to (approximately) follow the true class-distribution, and hence it can take advantage of a more balanced class distribution of unlabeled dataset.

To further investigate this phenomenon, we also evaluate algorithms for unlabeled dataset with reversely ordered class-distribution, i.e., $M_1 \leq \cdots \leq M_K$ and $M_k = M_1 \cdot \gamma_u^{-\frac{k-1}{K-1}}$ for $\gamma_u = 100$, denoted by "$\gamma_u = 100$ (reversed)" in Table 2. As expected, SSL algorithms fail as they provide wrong pseudo-labels to the most of unlabeled data (which are majority in unlabelded data while minority in labeled data). In contrast, DARP successfully refines theses pseudo-labels and significantly improves baselines as in prior experiments. For example, DARP exhibits 19.5%/22.5% relative error reductions of bACC/GM compared to the second-best method ReMixMatch* under $\gamma_u = 100$ (reversed).

**CIFAR-100 and STL-10.** We also present experimental results on CIFAR-100 and STL-10 datasets. In the case of CIFAR-100, we construct its "synthetically long-tailed" variants as done in Section 4.1, and assume that labeled and unlabeled datasets have the same class distribution, i.e., $\gamma_l = \gamma_u$. Since STL-10 also has a balanced labeled dataset with $N_k = 500$ for $k = 1, \ldots, 10$, we construct "synthetically long-tailed" variants with $N_1 = 450$. We fully use a given unlabeled data in STL-10 with $M = 100,000$, whose class distribution is unknown. Hence, in case of STL-10, labeled and unlabeled datasets may not have the same class distribution (i.e., $\gamma_l \neq \gamma_u$) and we estimate $\{M_k\}_{k=1}^K$ for DARP as previously conducted for Table 2. Table 3 summarizes the performance of baseline algorithms and DARP for learning both CIFAR-100 and STL-10. One can verify that DARP consistently improves the applied SSL algorithms for both datasets. It is also noticeable that the gain from DARP is significant on STL-10. This is because the mismatch of labeled and unlabeled datasets is significant in STL-10 as the given unlabeled dataset is usually known to be closed to the uniform distribution [5, 11]. Consequently, this result shows the importance of consideration of distribution mismatch between labeled and unlabeled data again, and the superiority of DARP.

### 4.3 Detailed analysis on DARP

**Comparison with other distribution matching.** The main motivation of DARP is to correct bias in pseudo-labels by matching the class distribution of pseudo-labels and true class distribution of unlabeled data. In this aspect, we compare DARP with the other distribution matching algorithms [2, 4], which are originally proposed under the balanced labeled/unlabeled class distributions, but applicable to any imbalanced settings. [2] directly adds the KL divergence loss as a regularizer where the class distribution of pseudo-labels is approximated within a mini-batch for optimizing loss. [4] proposes the "distribution alignment" procedure, which re-scales pseudo-labels to match

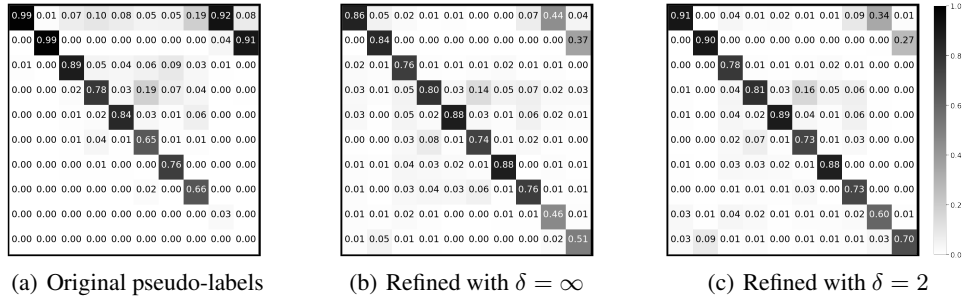

|(a) Original pseudo-labels|(b) Refined with $\delta = \infty$|(c) Refined with $\delta = 2$|

Figure 2: Confusion matrices of pseudo-labels on CIFAR-10 under $\gamma_l = 100, \gamma_u = 1$ using MixMatch [5]. (a) is from the original pseudo-labels. (b) and (c) are from refined pseudo-labels via DARP under $\delta = \infty$ and $\delta = 2$, i.e., without/with removing small entries, respectively.

Table 4: Comparison of classification performance (bACC/GM) on CIFAR-10 across different distribution matching methods applied to ReMixMatch [4] under $\gamma_l = 100$.

| Algorithm | $\gamma_u = 1$ | $\gamma_u = 50$ | $\gamma_u = 100$ | $\gamma_u = 100$ (reversed) |
|---|---|---|---|---|
| [2] | 81.4 / 80.5 | 76.0 / 72.7 | 72.5 / 67.8 | 72.9 / 67.1 |
| [4] | 85.0 / 84.3 | 77.0 / 74.7 | 73.8 / 69.5 | 75.3 / 72.3 |
| DARP | **89.7 / 89.4** | **77.4 / 75.0** | **75.8 / 72.6** | **80.1 / 78.5** |

the class distribution and then re-normalize them to satisfy the probability constraint. However, both methods cannot guarantee the exact distribution matching. Table 4 clearly shows that DARP provides larger performance gains compared to other algorithms. Note that we use the same target distribution $\{M_k\}_{k=1}^{K}$ for all algorithms. Here, one can observe that the performance gap between DARP and other algorithms becomes significant as the mismatch of labeled and unlabeled datasets becomes severe. This is because DARP exactly matches the distributions, while other algorithms do not.

**Quality of refined pseudo-labels.** We further evaluate DARP by measuring the error from refined pseudo-labels using underlying true labels hidden to DARP. For this, we use the model trained on CIFAR-10 under $\gamma_l = 100, \gamma_u = 1$ using MixMatch [5], which originally suffers from the biased pseudo-labels (see Figure 1(b)). Figure 2 visualizes the confusion matrix $C^{\texttt{unlabeled}}$ of pseudo-labels and refined pseudo-labels where $C^{\texttt{unlabeled}}$ is defined in Section 3.3. As expected, the original pseudo-labels are highly biased toward majority classes of the labeled dataset, i.e., small class indices. On the other hand, refined pseudo-labels are more likely to be unbiased compared to the original one, and the quality of pseudo-labels is significantly improved, especially in minority classes. In particular, for small $\delta$, the confusion matrix is less biased, while it is more biased for large $\delta$. We explain such an observation as follows: given $\delta$, the number of possible $k$-th nonzero entries in all refined pseudo-labels is restricted by $\delta M_k$. Here, large $\delta$ allows more freedom in the choice of entries of the refined pseudo-labels, which would result in a smaller distortion from the original biased pseudo-labels (see our optimization objective (1)). Hence, for large $\delta$, the resulting refined pseudo-labels are likely to preserve the original pseudo-labels' properties, including its bias. Besides, small $\delta$ regularizes the entries of refined pseudo-labels and hence, reduces the bias of the original pseudo-labels. The effect of other components of DARP is presented in the supplementary material.

## 5 Conclusion

In this paper, we propose a simple and effective method to refine pseudo-labels for semi-supervised learning (SSL) under assuming class-imbalanced training distributions. Our main idea is to refine the biased pseudo-labels (generated by an SSL algorithm) so that (a) their distribution match to the true class distribution and (b) they still preserve the information of the original pseudo-label as much as possible. To further increase the quality of the refined pseudo-labels, we suggest to remove some noisy entries in the original pseudo-labels. Our method is quite easy-to-use to be adapted to any SSL algorithms. The class-imbalanced SSL scenarios are under-explored in the literature, and we think our work can be a strong guideline when other researchers pursue these tasks in the future.

## Broader Impact

In this paper, we first identify that current state-of-the-art semi-supervised learning (SSL) algorithms can suffer from the class-imbalanced distribution of training data due to the biased prediction toward majority classes. Then, we propose a Distribution Aligning Refinery of Pseudo-label (DARP), which corrects such biased pseudo-labels from any SSL algorithms by solving the proposed optimization based on the knowledge of underlying distribution.

While this paper focused on the ordinary classification problem under class-imbalanced distribution, we expect that our work can contribute in a broader way, such as resolving the undesirable bias of deep neural networks (DNNs). Recently, it has been revealed that DNNs are often misled to exploit unintended correlation when the dataset is highly biased, although they achieve state-of-the-art performances on many tasks in artificial intelligence. The lack of de-biased samples might incur this phenomenon, one can address this by gathering such data without labels. However, in this way, the situation can deteriorate, i.e., the bias of DNNs can be severed, as we have identified in this work since the existing SSL algorithms mainly rely on the current prediction. However, by leveraging the prior knowledge, our method provides a safe way for utilizing the unlabeled data in this scenario, so that one can get desired de-biased models.

Simultaneously, our work reveals the vulnerability of recent state-of-the-art semi-supervised learning (SSL) algorithms under realistic scenarios. After [30] points out the limitation of current SSL algorithms, especially about the existence of out-of-distribution samples within unlabeled dataset, this scenario is recently considered by many researchers for stepping forward to the real-world application [10, 29]. However, as we have identified in this work, given class-imbalanced distribution can also be problematic. Even this scenario frequently occurs in the real-world, this direction is relatively under-explored so far [38]. Hence, we expect our work can encourage future researchers to focus on this crucial yet unnoticed direction for the application of semi-supervised learning in the real world.

## Acknowledgements

This work was supported by Samsung Advanced Institute of Technology (SAIT). This work was partly supported by Institute of Information & Communications Technology Planning & Evaluation (IITP) grant funded by the Korea government (MSIT) (No.2019-0-00075, Artificial Intelligence Graduate School Program (KAIST)).

## Footnotes

[1]The number of training samples of a class (in both labeled and unlabeled datasets) is up to 150 times smaller than that of another class.

[2] $\mathtt{Solve}_{Z \geq 0}(f(Z))$ returns $Z \geq 0$ such that $f(Z) = 0$. We utilize the Newton's method [6, 14] for $\mathtt{Solve}_{Z \geq 0}(f(Z))$ in all experiments in this paper.

[3] Consequently, when we apply DARP to a SSL algorithm, the additional running time incurred by DARP is at most 20% of that of the vanilla SSL algorithm in our experiments.

[4]We split the labeled dataset as $\mathcal{D}^{\texttt{labeled}} = \mathcal{D}^{\texttt{est}} \cup \mathcal{D}^{\texttt{train}}$ where $\mathcal{D}^{\texttt{est}} \cap \mathcal{D}^{\texttt{train}} = \emptyset$. Then, we train another classifier $g : \mathbb{R}^d \to [0, 1]^K$ using $\mathcal{D}^{\texttt{train}}$, and obtain the confusion matrix $C^{\texttt{est}}$ using $g$ on $\mathcal{D}^{\texttt{est}}$. In our experiments, we construct $\mathcal{D}^{\texttt{est}}$ by taking 10 samples for each class and train $g$ using a vanilla scheme. After that, we train a classifier $f$ by fully using $\mathcal{D}^{\texttt{labeled}}$ for training with $C^{\texttt{est}}$ as the estimation of $C^{\texttt{unlabeled}}$.

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
