[Supplementary Material]

# Supplementary Material:

## Distribution Aligning Refinery of Pseudo-label for Imbalanced Semi-supervised Learning

## A  Proof of Theorem 1

In this section, we present the formal proof of Theorem 1. To this end, we interpret DARP as a coordinate ascent algorithm of the Lagrangian dual of its original objective (1), and discuss the necessary and sufficient condition of correct convergence of DARP, i.e., convergence to the optimal solution of (1).

More generally, we consider the following problem: For $A \in \mathbb{R}_{\geq 0}^{n \times m}$, $r \in \mathbb{R}_{\geq 0}^n$, $c \in \mathbb{R}_{\geq 0}^m$ and $w \in \mathbb{R}_{\geq 0}^m$ such that $\|r\|_1 = \|c\|_1$, find the solution $M$ of the below convex optimization problem.

$$\underset{M \in \mathbb{R}_{\geq 0}^{n \times m}}{\text{minimize}} \qquad \sum_{i=1}^{n} \sum_{j=1}^{m} w_i M_{ij} \log \frac{M_{ij}}{A_{ij}} \tag{3}$$
$$\text{subject to} \qquad M \mathbf{1}_m = r \quad M^\top \mathbf{1}_n = c$$

where $\mathbf{1}_n$ denotes the $n$-dimensional vector consisting of ones. We note that the above problem is well-known in information theory as the I-projection [12] when $w_i = 1 \;\; \forall i$. Now, we will show that DARP is indeed a coordinate ascent algorithm for the dual of the above optimization. To this end, we formulate the Lagrangian dual of (3).

**Lemma 1.** *The Lagrangian dual of* (3) *is given by*

$$\underset{\lambda \in \mathbb{R}^n, \nu \in \mathbb{R}^m}{\text{maximize}} \qquad g(\lambda, \nu) = -\sum_{i=1}^{n} \sum_{j=1}^{m} w_i A_{ij} e^{-\frac{w_i + \lambda_i + \nu_j}{w_i}} - \sum_{i=1}^{n} \lambda_i r_i - \sum_{j=1}^{m} \nu_j c_j \tag{4}$$

*In addition, the optimal objective value of* (3) *is equivalent to that of* (4)*, i.e., the strong duality holds. Here, the primal optimal solution $M$ of* (3) *and the dual optimal solutions $\lambda, \nu$ of* (4) *satisfies that*

$$M_{ij} = A_{ij} e^{-\frac{w_i + \lambda_i + \nu_j}{w_i}} \;\; \forall i,j \tag{5}$$

We provide the formal proof of Lemma 1 in Section B.1 for the completeness. Apparently, (4) is a concave optimization problem and solving (4) directly provides the optimal solution of (3) due to the last statement of Lemma 1. To show the connection between DARP and (4), we introduce the below lemma. We present the formal proof of the below lemma in Section B.2.

**Lemma 2.** *A pair $(\lambda, \nu)$ is the unique optimal solution of* (4) *if and only if the below equalities hold:*

$$\sum_{j=1}^{m} A_{ij} e^{-\frac{w_i + \lambda_i + \nu_j}{w_i}} = r_i, \qquad \sum_{i=1}^{n} A_{ij} e^{-\frac{w_i + \lambda_i + \nu_j}{w_i}} = c_j. \tag{6}$$

*In addition, for the coordinate ascent updates $x^*, y^*$ for* (4) *defined by*

$$\lambda^* := \underset{\lambda \in \mathbb{R}_{>0}^n}{\arg\max}\, g(\lambda, \nu), \qquad \nu^* := \underset{\nu \in \mathbb{R}_{>0}^m}{\arg\max}\, g(\lambda, \nu). \tag{7}$$

*find the solutions of the former and latter equality of* (6)*, respectively.*

By substituting $\alpha_i = e^{-\frac{\lambda_i + w_i}{w_i}}, \beta_j = e^{-\nu_j}$, one can verify that DARP is exactly a coordinate ascent (7). Therefore, we conclude that DARP indeed optimizes the dual of (3) and it finds the unique optimum $M$ once it converges due to the last statement of Lemma 1. In particular, its correct convergence is guaranteed if the unique critical point of $g(\lambda, \nu)$ exists, i.e., there exists $\lambda, \nu$ satisfying (6).

# B Proofs of technical Lemmas

## B.1 Proof of Lemma 1

The Lagrangian dual function $g(\lambda, \nu)$ is defined by

$$g(\lambda, \nu) := \inf_{M \in \mathbb{R}_{\geq 0}^{n \times m}} \sum_{i=1}^{n} \sum_{j=1}^{m} w_i M_{ij} \log \frac{M_{ij}}{A_{ij}} + \sum_{i=1}^{n} \lambda_i \left( \sum_{j=1}^{m} M_{ij} - r_i \right) + \sum_{j=1}^{m} \nu_j \left( \sum_{i=1}^{n} M_{ij} - c_j \right). \tag{8}$$

Since $g$ is convex with respect to $M$, the desired $M$ can be computed by finding the critical point, i.e., $M$ minimizes the above objective if and only if each $M_{ij}$ satisfies

$$w_i \log \frac{M_{ij}}{A_{ij}} + w_i + \lambda_i + \nu_j = 0 \quad \Leftrightarrow \quad M_{ij} = A_{ij} e^{-\frac{w_i + \lambda_i + \nu_j}{w_i}}. \tag{9}$$

Using the above identity, we reformulate (8) as

$$g(\lambda, \nu) = \sum_{i=1}^{n} \sum_{j=1}^{m} w_i A_{ij} e^{-\frac{w_i + \lambda_i + \nu_j}{w_i}} \left( -\frac{w_i + \lambda_i + \nu_j}{w_i} \right) \tag{10}$$

$$+ \sum_{i=1}^{n} \lambda_i \left( \sum_{j=1}^{m} A_{ij} e^{-\frac{w_i + \lambda_i + \nu_j}{w_i}} - r_i \right) + \sum_{j=1}^{m} \nu_j \left( \sum_{i=1}^{n} A_{ij} e^{-\frac{w_i + \lambda_i + \nu_j}{w_i}} - c_j \right)$$

$$= -\sum_{i=1}^{n} \sum_{j=1}^{m} w_i A_{ij} e^{-\frac{w_i + \lambda_i + \nu_j}{w_i}} - \sum_{i=1}^{n} \lambda_i r_i - \sum_{j=1}^{m} \nu_j c_j$$

Hence, the Langrangian dual of (3) is given by (4). Since the optimization (3) satisfies the Slater's condition, i.e., there exists a feasible solution of (3) [38, 6], the strong duality holds. Finally, the optimal solution $M$ of (3) satisfies $M_{ij} = A_{ij} e^{-\frac{w_i + \lambda_i + \nu_j}{w_i}}$ due to (9). This completes the proof of Lemma 1

## B.2 Proof of Lemma 2

It is straightforward to derive that

$$\frac{g(\lambda, \nu)}{\partial \lambda_i} = \sum_{j=1}^{m} A_{ij} e^{-\frac{w_i + \lambda_i + \nu_j}{w_i}} - r_i, \qquad \frac{g(\lambda, \nu)}{\partial \nu_j} = \sum_{i=1}^{n} A_{ij} e^{-\frac{w_i + \lambda_i + \nu_j}{w_i}} - c_i.$$

Since $g(\lambda, \nu)$ is concave for $\lambda$ (or $\nu$) given $\nu$ (or $\lambda$), $\lambda^*$ (or $\nu^*$) is a coordinate ascent update if and only if $\frac{g(\lambda, \nu)}{\partial \lambda} = \mathbf{0}_n$ (or $\frac{g(\lambda, \nu)}{\partial \nu} = \mathbf{0}_m$). This directly implies the second statement of Lemma 2 and completes the proof of Lemma 2.

# C Experimental results on SUN397

Table 5: Comparison of classification performance (bACC/GM) on SUN397. SSL denotes semi-supervised learning and RB denotes re-balancing. The numbers in brackets below the gray rows are relative test error gains from DARP, compared to applied baseline SSL algorithms, respectively. The best results are indicated in bold.

| Method | SSL | RB | $\gamma \approx 46$ |
|---|---|---|---|
| Vanilla | - | - | $38.3_{\pm0.05}$ / $29.9_{\pm0.08}$ |
| cRT | - | ✓ | $39.3_{\pm0.21}$ / $33.7_{\pm0.37}$ |
| FixMatch | ✓ | - | $44.9_{\pm0.11}$ / $35.7_{\pm0.66}$ |
| FixMatch + DARP | ✓ | - | $\mathbf{45.5}_{\pm0.32}$ / $\mathbf{37.5}_{\pm0.04}$ |
| | | | (-1.09% / -2.80%) |

In this section, we further verify the effectiveness of DARP on the real-world imbalanced dataset, SUN-397 [44]. SUN-397 is a dataset for a scene categorization and originally consists of 108,754 RGB images labeled with 397 classes. Since the dataset itself does not provide any separated split for testing, we first hold-out 50 samples per each class for testing. After that, we artificially construct the labeled and unlabeled dataset using the remaining dataset under $\frac{M_k}{N_k} = 2$. See Figure 3 for the class distribution of constructed dataset. With this dataset, we compare the models trained under 4 representative

Figure 3: Class distribution of labeled and unlabeled dataset of SUN-397.

algorithms: vanilla, classifier re-training (cRT) [23], FixMatch [39] and FixMatch+DARP. Table 5 summarizes the experimental results. Here, DARP surpasses all the baseline algorithms, and this result shows the extensibility of DARP toward the real-world dataset.

**Training details.** For pre-processing, we randomly crop and rescale to $224 \times 224$ size all labeled and unlabeled training images before applying augmentation. Following the experimental setups in [39], our batch contains 128 labeled data and 640 unlabeled data on each step. Here, we use standard ResNet-34 [19] model and train it for 300 epochs of unlabeled data using SGD optimizer with momentum 0.9. We utilize linear learning rate warmup for the first 5 epochs until it reaches an initial value of $0.4$. Then, we decay the learning rate at epochs 60, 120, 160, and 200 epochs by dividing it by 10. For applying FixMatch [39], we use unlabeled loss weight $\lambda_u = 5$ and confidence threshold $\tau = 0.7$. We utilize Exponential moving average technique with decay 0.9 and use RandAugment with random magnitude [13] for strong augmentation and random horizontal flip for weak augmentation.

# D  Combination of re-balancing and semi-supervised learning

Table 6: Comparison of classification performance (bACC/GM) on CIFAR-10 under three different class-imbalance ratio $\gamma = \gamma_l = \gamma_u$. SSL denotes semi-supervised learning and RB denotes re-balancing. The numbers in brackets below the gray rows are relative test error gains from DARP, compared to applied baseline SSL algorithms, respectively. The best results are indicated in bold.

| | | | CIFAR-10 | | |
|---|---|---|---|---|---|
| Algorithm | SSL | RB | $\gamma = 50$ | $\gamma = 100$ | $\gamma = 150$ |
| Vanilla | - | - | $65.2_{\pm 0.05}$ / $61.1_{\pm 0.09}$ | $58.8_{\pm 0.13}$ / $51.0_{\pm 0.11}$ | $55.6_{\pm 0.43}$ / $44.0_{\pm 0.98}$ |
| ReMixMatch [4] | ✓ | - | $81.5_{\pm 0.26}$ / $80.2_{\pm 0.32}$ | $73.8_{\pm 0.38}$ / $69.5_{\pm 0.84}$ | $69.9_{\pm 0.47}$ / $62.5_{\pm 0.35}$ |
| ReMixMatch + DARP | ✓ | - | $82.1_{\pm 0.14}$ / $80.8_{\pm 0.09}$ | $75.8_{\pm 0.09}$ / $72.6_{\pm 0.24}$ | $71.0_{\pm 0.27}$ / $64.5_{\pm 0.68}$ |
| | | | (-3.45% / -3.52%) | (-7.84% / -10.2%) | (-3.60% / -5.19%) |
| ReMixMatch + cRT | ✓ | ✓ | $86.8_{\pm 0.50}$ / $86.5_{\pm 0.49}$ | $81.4_{\pm 0.41}$ / $80.7_{\pm 0.45}$ | $78.9_{\pm 0.84}$ / $77.8_{\pm 0.94}$ |
| ReMixMatch + DARP + cRT | ✓ | ✓ | $\mathbf{87.3}_{\pm 0.16}$ / $\mathbf{87.0}_{\pm 0.11}$ | $\mathbf{83.5}_{\pm 0.07}$ / $\mathbf{83.1}_{\pm 0.09}$ | $\mathbf{79.7}_{\pm 0.54}$ / $\mathbf{78.9}_{\pm 0.49}$ |
| | | | (-4.33% / -3.63%) | (-11.3% / -12.2%) | (-3.61% / -5.12%) |

Meanwhile DARP outperforms all the baselines, it could be further improved by combining with RB algorithms. To verify this, we examine DARP and baseline SSL algorithm, ReMixMatch [4], by combining with the classifier re-training (cRT) algorithm [23], which is a recently introduced state-of-the-art RB algorithm. These algorithms are denoted by "ReMixMatch + cRT" and "ReMixMatch + DARP + cRT", respectively, and the detailed implementation can be found in Section E. Table 6 summarizes the performance of ReMixMatch with/without DARP combined with cRT. One can observe that combining with cRT improves both ReMixMatch and DARP significantly, but DARP still outperforms ReMixMatch under cRT. For example, DARP exhibits 11.3%/12.2% relative error gains of bACC/GM compared to ReMixMatch when $\gamma = \gamma_l = \gamma_u = 100$.

# E  Implementation details

All experiments are conducted with Wide ResNet-28-2 [33] and it is trained with batch size 64 for $2.5 \times 10^5$ training iterations except Section C. For training with semi-supervised learning algorithms, we use Adam optimizer [25] with a learning rate $2 \times 10^{-3}$. For the hyperparameters of Adam, we use $\beta_1 = 0.9$, $\beta_2 = 0.999$ and $\varepsilon = 10^{-8}$ which is a default choice of [25]. To evaluate models trained by semi-supervised learning algorithms, we use an exponential moving average (EMA) of its parameters with a decay rate $0.999$ and apply weight decay of $4 \times 10^{-4}$ at each update following [5].[5] For training with re-balancing algorithms, we use SGD with a learning rate $0.1$, momentum $0.9$ and weight decay $5 \times 10^{-4}$. The learning rate of SGD decays by $0.01$ at the time step $80\%$ and $90\%$ of the total iterations following [8]. For all algorithms, we evaluate the model on the test dataset for each 500 iterations and report the average test accuracy of the last 20 evaluations following [5].

**DARP.** We apply DARP procedure for each 10 iterations with fixed hyper-parameters $\delta = 2$ and $T = 10$, which is empirically enough for the convergence of DARP. To show its convergence empirically, we directly measure the distribution mismatch as $\frac{1}{M} \sum_{k=1}^{K} |\tilde{M}_k - M_k|$ for each iteration, where $\tilde{M}_k$ implies the resulting class distribution of refined pseudo-labels. For additional comparison, we also report the results from distribution alignment (DA) term of ReMixMatch [4]. We use the model trained using MixMatch [5] under 3 cases: (1) $\gamma_l = 100, \gamma_u = 1$, (2) $\gamma = \gamma_l = \gamma_u = 100$ (reverse) and (3) $\gamma = 100$. Figure 4 shows that $T = 10$ is enough for the convergence and DARP's superiority compared to DA. The effect of $\delta$ is presented in Section 4.3. Since the estimation is not accurate at the early stage of training, we are not using DARP until the first $40\%$ of iterations.

Figure 4: Distribution matching via DARP for each iteration.

**VAT, Mean-Teacher and MixMatch.** In the experiments, we use VAT [31], Mean-Teacher [40] and MixMatch [5] as the baselines for semi-supervised learning methods. Since these methods have some hyperparameters, we use the same values for them as used in [5]. For VAT, the consistency coefficient $\lambda_u$ is set to $0.3$, the norm constraint for adversarial perturbation $\epsilon$ is set to $1.0$ and the norm of initial perturbation $\xi$ is set to $1.0 \times 10^{-6}$. For Mean-Teacher, the consistency coefficient $\lambda_u$ is set to $50$ and the EMA model used for the evaluation is reused for the consistency regularization. Following [40], we ramped up the consistency coefficient starting from 0 to $\lambda_u$ using a sigmoid schedule so that it achieves the maximum value at $1.0 \times 10^5$ iterations for both methods. For MixMatch, we set temperature $T$ as $0.5$, the number of augmentation $K$ as $2$, the parameter for beta distribution $\alpha, \beta$ as $0.75$ and the consistency coefficient $\lambda_u$ as $75$. During the entire iterations, the consistency coefficient is linearly increased to $\lambda_u$ started from $0$.

**ReMixMatch.** For ReMixMatch [4], we use $K = 2$ for the number of augmentations to balance the improvement from an augmentation anchoring and a computational cost. Also, we use RandAugment [13] as a strong augmentation which is shown to be substitutable with CTAugment [39]. For the other hyperparameters, we use the same values used in the original paper.

**FixMatch.** For FixMatch [39], we use $\mu = 2$ for the ratio of unlabeled data $\mu$ and Adam optimizer as described in Section 4.1 to fairly compare it with ReMixMatch. For the other hyperparameters, $\lambda_u$ used for balancing the loss from labeled and unlabeled data and $\tau$ used as a threshold for pseudo-labeling [28], we use the same values with the original paper, $\lambda_u = 1$ and $\tau = 0.95$.

**cRT.** To re-balance the network trained under the imbalanced class distribution, we use the classifier re-training (cRT) scheme [23]. After training the network regardless of the use of unlabeled data without considering the imbalance, we re-initialize the linear classifier of the network and only re-train it for $1.0 \times 10^4$ iterations by freezing other parameters. We re-balance the training objective by re-weighting the given loss [24] instead of re-sampling. Also, we re-train the classifier using only the labeled dataset.

## Footnotes

[5]We do not use the exponential moving average for evaluating "re-balancing" algorithms as it hurts the performance.