[Reviews · NeurIPS 2020]

Review 1

Summary and Contributions: This paper proposes a simple technique DARP to refine the biased pseudo-labels for imbalanced semi-supervised learning (SLL), and DARP is applicable to many existing SSL methods. **The authors addressed my questions. The experiments do show promising results, but I think the theoretical gounding is a little weak. **

Strengths: 1. The proposed method DARP is simple and effective to improve the performance of existing SLL algorithms for handling imbalanced SLL problem. 2. The experiments are well-designed.

Weaknesses: 1. Algorithm 1 is proposed to solve Eq. (1), but the derivation process is not very clear and the notations are somewhat confused (e.g. X denotes the input, does it mean the same thing in Algorithm 1?). 2. It is declared in Sec. 3.2 that "DARP increases at most 20% of the overall training time of an existing SSL scheme". I don't quite understand. 3. The experiments show that DARP could improve the performance of the SLL methods. But I observe that the results of SLL methods which are not specifically designed for imbalanced SLL (VAT, Mean-Teacher...) are comparable with that of imbalanced SLL methods. This may seem a little strange, so I wonder the reason.

Correctness: The technique appears to be correct.

Clarity: The paper is readable.

Relation to Prior Work: The paper deals with a valuable problem (imbalanced SLL), reviews the previous works and compares with them. While there is still room for improvements in the motivation because the difference and connection with previous works are not very clear.

Reproducibility: Yes

Additional Feedback:


Review 2

Summary and Contributions: Distribution Aligning Refinery of Pseudo-label (DARP) For semi-supervised learning (SSL), DARP is proposed to match the pseudo-labels with the underlying class distribution of the unlabeled data. The objective function is to minimize the KL divergence of the "aligned" pseudo-labels with the original pseudo-labels subject to the constraints that the "aligned" pseudo-labels are consistent with desired class/label distribution for the unlabeled data. To speed up the process, DARP uses a coordinate ascent algorithm for the Largrangian dual of the objective function. The evaluation was conducted with the CIFAR10 dataset with various artificially degrees of imbalance. DARP was used with a few existing algorithms for imbalanced SSL. One scenario is the unlabeled data has the same class distribution as the labeled data. A second scenario is when the two class distributions are different. In the second scenario, they estimated the class distribution of the unlabeled using essentially a validation set from the training data. DARP was also compared with two existing distribution matching methods. The results are generally favorable for DARP.

Strengths: The main strength of the DARP algorithm is to align pseudo-labels with the desirable class distribution in the unlabeled data for semi-supervised learning with imbalanced labeled data. Empirical results are favorable compared against existing techniques.

Weaknesses: The main weakness is the justification for the proposed method for estimating the desired/actual class distribution of the unlabeled data, which is not known. Another weakness is that only the CIFAR10 dataset was used for evaluation. [After the author feedback: while in the supplementary materials, Section F has justification and Sections C&D have results from additional data sets, summarizing some insights of the justification and key results from other datasets in the main text would be beneficial. ]

Correctness: The methods and empirical methodology are reasonable.

Clarity: The paper is generally well written. For the desired/actual class distribution for the unlabeled data (which is not known), the reasoning for the proposed estimation technique could be expanded in the main paper. This is quite important for the DARP algorithm. Also, expanding on why setting small probabilities to zero enhance the quality of pseudo labels would be beneficial.

Relation to Prior Work: Discussion with prior work is reasonable.

Reproducibility: Yes

Additional Feedback: line 32 and Figure 1b: the ratio seems to be less than 4 (100/25) in Figure 1b rather than 1046 on line 32. line 119: true distribution is assumed from the labelled training data? eq 1: what is \hat{y}_m? Looks like the refined pseudo labels line 134: why removing small entries enhances the quality of pseudo-labels? Table 1: test error gains are negative, might be simple to say test error reduction is positive line 203: test error gains are positive? (not consistent with Table 1) line 212: What is the justification for: "we approximate C^{unlabeled} using some small subset of the labeled dataset, which is not used for the training until the confusion matrix is estimated."


Review 3

Summary and Contributions: This paper focuses on semi-supervised learning under the class imbalance problem. They propose a class distribution matching method to refine the pseudo-labels for unlabeled data. Specifically, they try to refine the biased pseudo-labels distribution to match the true distribution of unlabeled data. Experiment results are reported on CIFAR-10 data sets. However, the proposed method is based on an assumption that the true distribution of unlabeled data needs to be known which is not feasible in real-tasks. Moreover, more discussion of the distribution estimation method needs to be discussed. And more experiment results on other data sets need to be reported.

Strengths: 1) The paper focuses on semi-supervised learning under class imbalance. This is an important problem and it is not well studied. 2) They propose a distribution matching method to refine the biased pseudo-label distribution. However, the method needs to know the true distribution of unlabeled data which is not feasible in real applications. So the contribution is quite limited. 3) The paper is relevant to the NeurIPS community.

Weaknesses: 1) The authors claim that the reason for SSL can not work well is that they adopt biased pseudo-labels. However, most deep SSL methods are based on the smooth assumption and encourage the original data and the augmented data have similar predictions. Actually, they don't need to assign a pseudo-label to an unlabeled example explicitly. So I think the claim does not make sense. 2) The proposed method in Section 3 is based on the true class distribution of unlabeled data. This is not feasible in real applications. Although the authors give an estimation method in the experiment section, it is still a problem with the practicability of the proposal. More analysis about the estimation method need to be discussed, for example, the distance between the estimated distribution and the true distribution 3) All experiment results are conducted on CIFAR-10 datasets, experiments on more data sets should be reported to demonstrate the effectiveness of the proposal. **The authors have carefully addressed issues on 2) and 3) in the rebuttal**

Correctness: The proposed method is based on the true distribution of unlabeled data that is not available in real applications

Clarity: No. More analysis of the distribution estimation method needs to be discussed.

Relation to Prior Work: Yes.

Reproducibility: Yes

Additional Feedback:


Review 4

Summary and Contributions: Semi-supervised learning models trained on label-imbalanced datasets tend to output even more biased prediction and therefore perform badly under balanced testing criterion. To overcome the problem, this work proposes an approach to refine pseudo labels to meet the prior label distribution.

Strengths: - As far as I know, it is the first work to explore the label-imbalanced problem in training deep SSL models. - The proposed approach is general and can be adopted in any self-training SSL method. - Authors empirically show that the proposed method consistently improves deep self-training methods such as MixMatch, ReMixMatch, FixMatch.

Weaknesses: - The work fails to compare with simple baseline methods that adopt both sample reweighting and unlabeled data. For example, reweight both labeled and unlabeled samples by using the inverse of (pseudo) label frequency. - It would be better to provide theoretical and empirical analysis on the computational cost of Alg. 1. - It would be better to perform experiments on real-world imbalanced datasets to validate the proposed method.

Correctness: To my best knowledge, the method is technically sound.

Clarity: The paper is well organized and clearly written.

Relation to Prior Work: There are many SSL methods that use class distribution as prior knowledge to align the prediction distribution. Please check section 7 of [1] for a review. However, the paper fails to refer to these works and discuss their relationship. [1] Xiaojin Zhu. Semi-Supervised Learning Literature Survey. Computer Sciences TR-1530, 2008.

Reproducibility: Yes

Additional Feedback:

[Author Response · NeurIPS 2020]

We sincerely appreciate insightful comments and positive feedback from the reviewers: important problem (**R1**, **R3**),
simple and effective (**R1**, **R4**), and good experiments (**R1**, **R2**, **R4**). We respond to each comment one by one.

(**R2**, **R3**) **Experiments on datasets other than CIFAR-10.** In fact, the experimental results on CIFAR-100, STL-10,
and SUN-397 datasets are presented in Sections C and D of the supplementary material. In these results, DARP
consistently outperforms other baselines. We mention this in Line 148; however, we will make it clear in the final draft.

(**R1**) **Imbalanced SSL.** We first clarify that SSL algorithms + DARP are the *only* 'imbalanced' SSL algorithms in
our experiments, which consistently outperform the two types of baselines: re-balancing (RB) and semi-supervised
learning (SSL). Here, RB algorithms are designed for imbalanced classification, but they do *not* use the unlabeled
data. Conversely, SSL algorithms use the unlabeled data but they do *not* consider the class imbalance. We suspect
that **R1** uses "imbalanced SSL" for denoting RB algorithms: it is not surprising that performances of RB and SSL are
comparable as one of them is not superior to the other in general. We will make this point clear in the final draft.

(**R1**) **Derivation/confusion in Algorithm 1.** We first note that the formal derivation of Algorithm 1 is presented in
Sections A and B of the supplementary material. Here, we use $X, Y$ for denoting auxiliary variables for Algorithm 1
while $x$ denotes the input. However, to avoid the confusion, we will substitute $X, Y$ to $\alpha, \beta$ in the final draft.

(**R1**) **Unclear sentence.** Following the **R1**'s comment, we will clarify "DARP increases..." to "When we apply DARP
to the SSL algorithm, the additional running time incurred by DARP is at most 20% of the running time of the vanilla
SSL algorithm in our experiments." in the final draft. We thank **R1** for pointing out this.

(**R2**) **Discussion for the proposed estimation.** We provide the detailed discussion of our estimation scheme in Section
F of the supplementary material. The linear relationship of the confusion matrix with the models' prediction and the
true distribution of the unlabeled data is the key of the proposed estimation. We will move this to the main text.

(**R2**) **Additional feedback.** We thank **R2** for careful reading to improve the presentation of our paper. We feel sorry
for not providing our response to them due to the space limit. We will clarify all comments in the final draft.

(**R3**) **Pseudo-labels for SSL algorithms.** As **R3** mentioned, most SSL algorithms encourage the original data and the
augmented data to have similar predictions, typically by matching the model's prediction of the original data to some
target vector (e.g., the model's prediction of the augmented data). In our paper, the target vector is denoted by the
'pseudo-label', i.e., it is a soft-label and not necessarily one-hot. To clarify the **R3**'s confusion, we will emphasize our
definition of the pseudo-label in the final draft.

(**R3**) **True distribution of the unlabeled data.** We first clarify that the true distribution

of the unlabeled data is NOT necessarily required for DARP. As reported in Table 2 and
experimental results on STL-10 in Table 4 of the supplementary material, DARP "with-
out the true distribution" is super effective for the imbalanced SSL problem. This is beca-
use we estimate the true distribution via our estimation scheme. Our estimation scheme is
based on the simple linear relationship (see Section 4.2 and Section F of the supplement-
ary material for details) widely used in various problems, e.g., domain adaptation [L$^+$18]
and noisy labels [H$^+$18]. In particular, as illustrated in the right figure, our estimation scheme (blue) effectively
approximates the true distribution (green). Namely, DARP can refine pseudo-labels to have distribution close to the true
one while the distribution of pseudo-labels generated by FixMatch (red) is highly biased. We will clarify these points
and move the detailed discussion of the estimation scheme in the supplementary material (Section F) to the main text.

Finally, as mentioned in Line 194, the distribution of the unlabeled data can be also inferred from that of labeled data
without such estimation, if both data are sampled from the same distribution (arguably the most practical scenario).

[L$^+$18] Lipton et al., "Detecting and Correcting for Label Shift with Black Box Predictors", ICML, 2018.
[H$^+$18] Hendrycks et al., "Using Trusted Data to Train Deep Networks on Labels Corrupted by Severe Noise", NeurIPS, 2018.

(**R4**) **Computational cost.** The computational cost of Algorithm 1 is $O(KM)$ where $K, M$ denote the number of
classes and the number of the unlabeled data, respectively. In our experiments, DARP increases at most 20% of the
overall training time of the vanilla SSL algorithm. We will add a related discussion in the final draft.

(**R4**) **Fail to compare with simple baseline.** In fact, DARP has been compared with algorithms that **R4** suggested, i.e.,
utilizing both re-balancing and the unlabeled data (i.e., SSL), in Section E of the supplementary material. Here, DARP
also benefits by re-balancing since it only resolves pseudo-labels' bias toward the majority classes. We will move this
result to the experiment section in the final draft.

(**R4**) **Real-world imbalanced dataset.** Experimental results on the imbalanced real-world scene categorization dataset
(SUN-397) is presented in Section D of the supplementary material. We will clarify this in the final draft.

(**R4**) **Related work.** In Section 4.3, we compare our method with recent SSL algorithms using the class distribution as
prior knowledge and discuss them. Nevertheless, following the **R4**'s suggestion, we will add a discussion about these
algorithms and works in "Semi-Supervised Learning Literature Survey" to the related work section to the final draft.

[Meta-Review · NeurIPS 2020]

This paper proposes an approach to semi-supervised learning for imbalanced classes. It is indeed non-trivial to combine local/global/perturbation consistency-based semi-supervised methods and fully supervised methods for imbalanced classes---this paper may be the first work along this direction. The paper is quite general and can be applied on top of any pseudo-labeling-based semi-supervised methods. It first estimates the true class-prior probability and then updates/modifies the pseudo labels by pushing their class-prior probability with a constrained convex optimization. While in the beginning the reviewers had some concerns (mainly the clarity and too few datasets), the authors did a particularly good job in their rebuttal (showing that the class-prior probability can be estimated rather than must be given). Thus in the end, all of us have agreed to accept this paper for publication! PS, reviewers are all voluntary and serve for free, and they are busy and not asked to go through all the things in the supp file. The authors shouldn't use the supp as same as the main and then rely on that all reviewers will also review the appendices, because this will implicitly break the page limit and be extremely unfair to other papers. So if the authors think certain messages are important (appendix F and the experiments on more datasets), these things should definitely be moved to the main file at the submission time. A strange organization of the material hurts the clarity very very much; without clarity, it is meaningless to talk about the novelty or the significance, where I think the authors should really learn a lesson this time. Moreover, please be more discreet in using the confidential comments to ACs, since we are all voluntary and serve for free!